# A Fast Aerodynamic Model for Aircraft Multidisciplinary Design and Optimization Process

**Frédéric Moëns**

ONERA, Université Paris-Saclay, F92190 Meudon, France; frederic.moens@onera.fr

**Abstract:** A multidisciplinary design analysis and optimization process is developed at ONERA for the design of tube and wing and blended wing–body aircraft configurations. This process is composed of different disciplinary modules (geometry, propulsion, aerodynamics, structure, handling qualities and flight mission), and the overall process considers different fidelity levels for these modules at each step of the design process. This article describes the low-fidelity aerodynamic module used during the preliminary design optimization process. Analytical formulations retained for lift and drag components are presented in the first part. Then, the performances estimated by the aerodynamic module on some reference configurations are compared with both numerical and experimental data, showing a quite good agreement for both tube and wing and blended wing–body configurations not only for global performance but also for individual drag components.

**Keywords:** aircraft performance; analytical method; drag evaluation; drag components; blended wing–body





## 1. Introduction

Among the different options for more efficient aircraft configurations, the blended wing–body (BWB) seems to offer a very promising reduction in emissions ($CO_2$, NOx). However, such a configuration needs a complete, integrated design process as all the different disciplines (cabin arrangement, aerodynamics, structure, propulsion, flight dynamics, etc.) strongly interact altogether [1]. In order to explore, in depth, the potential benefits of such configurations, a multidisciplinary design analysis and optimization (MDAO) process dedicated to blended wing–body aircraft configurations has been developed since 2015 at ONERA [2–5]. The design workflow covers the range of tools from level 0 (L0), considering (semi-) empirical methods, over level 1 (L1), taking into account low-level physics-based methods, to level 2 and 3 (L2, L3), considering high-fidelity methods (Figure 1).

The modules implemented are easily exchangeable to adapt the workflow for specific configurations or a higher level of fidelity in specific domains if different methods are needed. This design tool evolves continually due to the enrichment of the disciplinary modules integrated or the addition of new disciplinary modules. The last version is composed of six main disciplinary modules: aircraft geometry, propulsion, aerodynamics, structure and weight, mission and performance and, finally, handling qualities. Additionally, some acoustic analysis can be performed offline of the multidisciplinary process for the selected configurations (Figure 2). Those modules are integrated in the NASA OpenM-DAO [6] framework to constitute the multidisciplinary design analysis and optimization (MDAO) process.

When the main characteristics of the aircraft are not fixed, a large number of configurations are evaluated during the exploratory phase of the overall aircraft design (OAD) process with a large variety of architectures and planforms. As an example, Figure 3 presents some BWB planforms evaluated for building up a multi-fidelity surrogate model for the first pre-design optimization loop, showing a large variety of architectures to be considered (from [4]). A first constraint for the aerodynamic module is to be able to manage 100% of these different architectures.

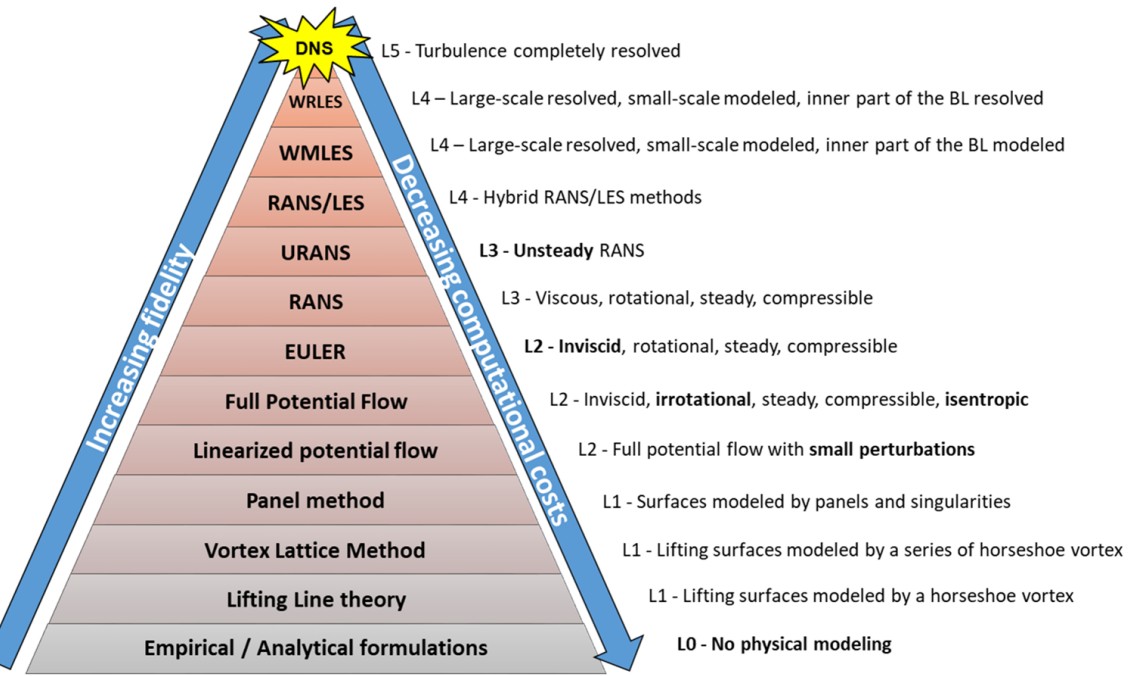

**Figure 1.** The different fidelity levels for flow physics modeling of the aerodynamic solvers.

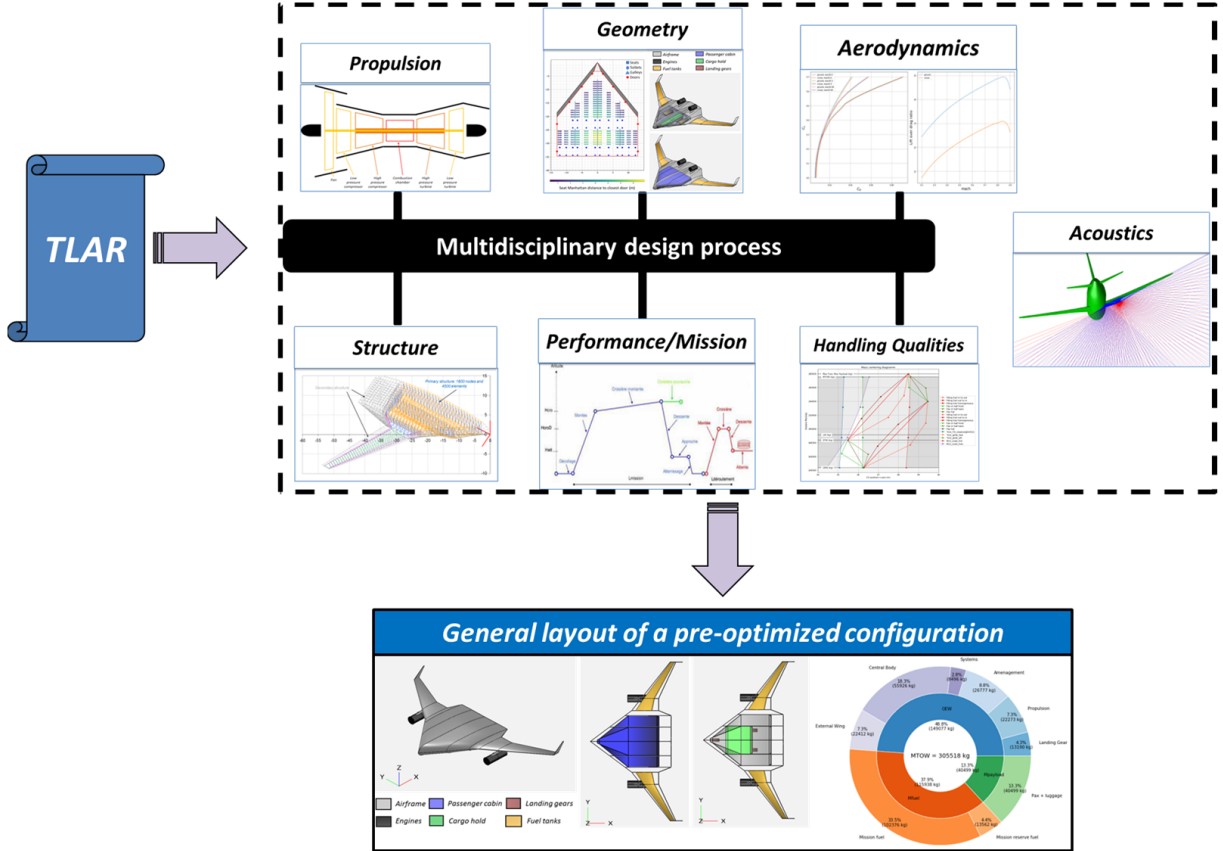

**Figure 2.** ONERA MDAO/OAD tool for BWB applications.

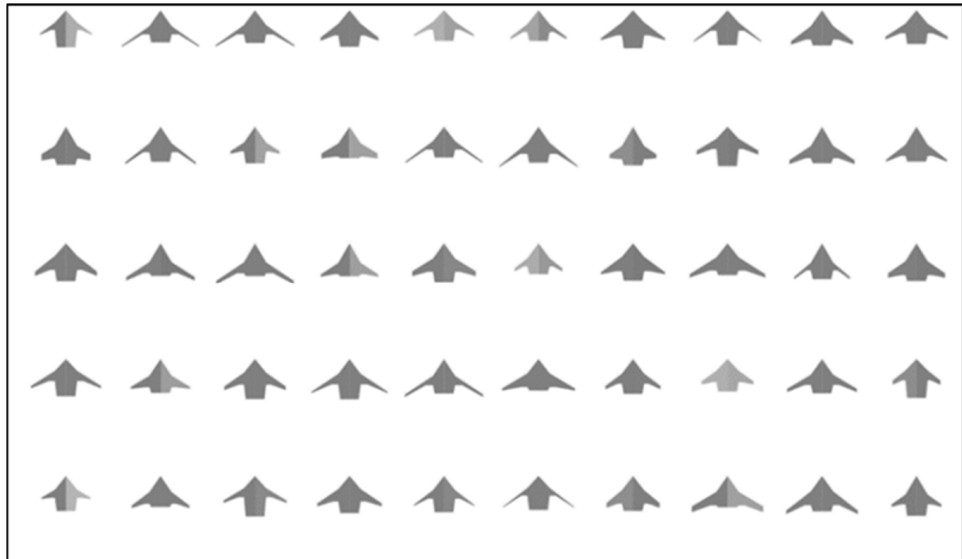

**Figure 3.** Extract of shapes considered for a DoE for a BWB design.

Then, for each configuration, the mission module generates an aerodynamic database. As an order of magnitude, these databases consider about 5000 aerodynamic flow conditions to be computed for <u>each</u> planform, which makes the use of an aerodynamic disciplinary module with a very fast computing time and a reliable performance level prediction mandatory.

Therefore, among the different numerical methods available for aerodynamic modeling, the ones belonging to the "low-fidelity" family (L0 or L1) are often used for OAD phases. For the present application, a dedicated L0 aerodynamic module based on analytical formulations derived from the theory or from the data analysis of past or present aircraft is developed for a fast evaluation of the aerodynamic performance for standard "Tube and Wing" (T&W), "Flying Wing" (FW) or "Blended Wing Body" (BWB) configurations for typical subsonic cruise flight conditions. This module considers only wing planform data, and no surface grid is needed.

The different formulations retained are described in the first part of the article. In the second part, the results issued from the module are compared to the numerical or experimental results of different aircraft configurations.

## 2. Standard Atmosphere Model

To characterize the evolution of the ambient static flow conditions throughout the atmosphere, the aerodynamic module integrates an analytic formulation of the International Standard Atmosphere (ISA) [7]. ISA is a mathematical model that divides the atmosphere into several layers (troposphere, stratosphere, mesosphere, etc.) that gives the evolution of the ambient pressure, temperature and density with the altitude. The flight altitudes considered in the module belong to the troposphere and the stratosphere. Within each layer or sublayer, the temperature is assumed to have a linear evolution with respect to the altitude, the pressure is calculated by the hydrostatic balance from Equation (1), and the air density is calculated assuming the air as a perfect gas from Equation (2):

$$\frac{dP}{dz} = -\rho g \tag{1}$$

$$\rho = \frac{P}{RT} \tag{2}$$

For the air considered as a perfect gas, there is R = 287.04 J kg$^{-1}$ K$^{-1}$, and the standard gravity constant is g = 9.80665 m/s$^2$.

Knowing the evolution of the temperature with the altitude (Figure 4) and considering some thermodynamic assumptions for the different layers, the integration of Equation (1) between the two altitudes that bound the given layer leads to some analytic formulations, as presented in Table 1.

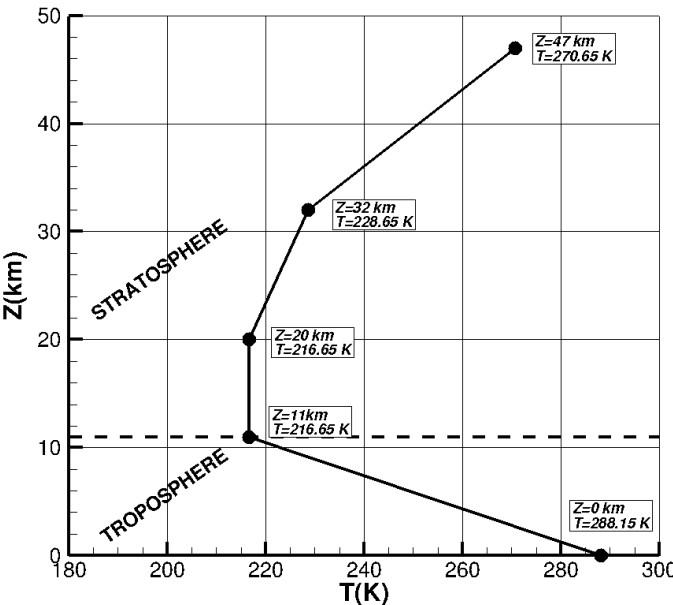

**Figure 4.** Evolution of the temperature with the altitude—ISA model.

**Table 1.** Analytic standard atmosphere model used.

| Layer | Altitude (km) | Temperature (K) | Pressure (Pa) |
|---|---|---|---|
| Troposphere | $Z \leq 11$ | $T = 288.15 - 6.5 \times Z$ | $P = 101325.0 \times \left( \frac{T}{288.15} \right)^{5.2561}$ |
| Stratosphere | $11 \leq Z \leq 20$ | $T = 216.65$ | $P = 22630.6 \times 10^{-\left( \frac{z-11}{14.596} \right)}$ |
| | $20 \leq Z \leq 32$ | $T = 216.65 - (Z - 20)$ | |

Finally, the air dynamic viscosity is computed using Sutherland's relation:

$$\mu = \mu_0 \left( \frac{T}{T_0} \right)^{\frac{3}{2}} \left( \frac{T_0 + 110.4}{T + 110.4} \right) \tag{3}$$

where $\mu_0 = 1.711 \ 10^{-5}$ kg/(ms) and $T_0 = 273.15$ K.

## 3. Aerodynamic Module

The aerodynamic module is dedicated to T&W, FW or BWB configurations in subsonic, attached flow conditions. It does not consider flow conditions with separation, buffet, high-lift configurations, supersonic flight and unconventional aircraft configurations, such as multiplane or boxed wing.

In the first stage, a reference wing is built based on geometrical characteristics such as the airfoil relative thickness, the leading-edge sweep angles, the local chord, etc., and reference geometrical parameters are then derived (reference surface, mean aerodynamic chord). For the performance evaluation, this wing is considered "aerodynamically optimized" with an elliptical span loading in order to have an estimation of the local lift coefficients.

In the second stage, the influence of other elements, such as the fuselage, winglets or nacelles, are considered as drag increments. Note that the span loading of the wing considering these elements is not computed by the module.

The module is suited to compare different wing planforms within a preliminary design loop. By construction, identical results will be obtained for two configurations with the same wing planform and the same airfoil thickness spanwise evolution. The optimization of geometrical details, such as the airfoil shape, camber or twist, is not possible with the use of this module and has, therefore, to be considered in the next step of the design process using more advanced L1 or L2 numerical methods.

The following chapters present the geometrical inputs used and the analytical formulations considered for the estimation of the aerodynamic performance (lift and drag).

### 3.1. Geometrical Inputs
### 3.1.1. Wing

The wing planform is defined by the use of $N_{AIRF}$ airfoil sections of $N_{WSEG}$ segments ($N_{AIRF} = N_{WSEG}+1$). By construction, the first airfoil is located at the symmetry plane $Y(1) = 0$, and the last section $Y(N_{AIRF})$ corresponds to the wing tip. For each wing segment, the leading-edge sweep angle, an index ($X_{Tra}$) for laminar computation and the airfoil technology factor used for wave drag calculations ("Korn factor", see chapter 3.5) are provided. Additionally, for each airfoil, the relative thickness (t/c) and an estimation of the maximum lift ($C_{l\,max}$) are provided. Figure 5 gives an example of data for a wing defined by five airfoil sections (or four wing segments).

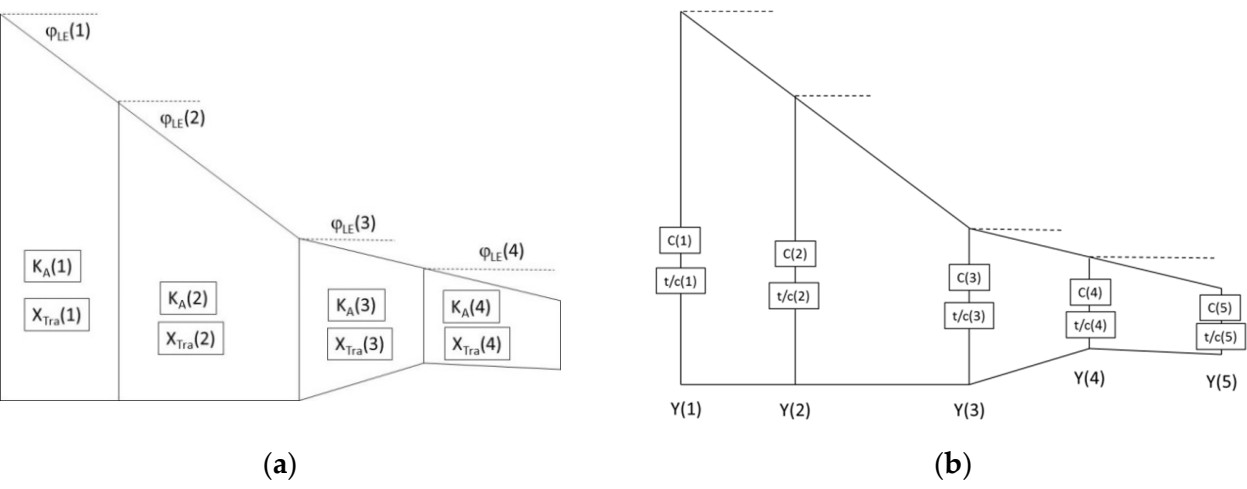

(**a**)  (**b**)

**Figure 5.** Geometrical data used for wing definition. (**a**) Geometrical data related to wing segments. (**b**) Geometrical data related to airfoil sections.

It should be noted that no detailed geometrical inputs for airfoils are used (shape, twist, camber).

Then, the wing span is divided into NDY subsections. For a given subsection, the different geometrical characteristics are obtained by a linear interpolation of the given parameter between two given wing sections of the segment. This allows the evolution of the leading-edge lines $X_{LE}(Y)$ and $Y_{LE}(Y)$, the chord law $C(Y)$ and the different sweep evolutions with wing span ($\varphi_0(Y)$, $\varphi_{25}(Y)$, $\varphi_{50}(Y)$) to be obtained.

The reference area considered for the aerodynamic coefficient calculations is the geometrical wing area obtained by the sum of the different trapezoid segments:

$$S_{REF} = S_{Wing} = \sum_{i=0}^{i=N_{WSEG}} \frac{[Y(i+1) - Y(i)] \times [C(i+1) + C(i)]}{2} \tag{4}$$

The wing geometrical aspect ratio $\lambda$ corresponds to:

$$\lambda = \frac{(2\,Y(N_{AIRF}))^2}{S_{Wing}} = \frac{b^2}{S_{Wing}} \tag{5}$$

The wing taper ratio $\varepsilon$ is:

$$\varepsilon = \frac{C(N_{AIRF})}{C(1)} \tag{6}$$

Finally, the value of the aerodynamic mean chord and its location on the half-wing can be obtained:

$$AMC = \frac{2}{S_{Ref}} \int_0^{b/2} C^2(y) dy \tag{7}$$

$$X_{AMC} = \frac{2}{S_{Ref}} \int_0^{b/2} X_{LE}(y) \times C(y) dy \tag{8}$$

$$Y_{AMC} = \frac{2}{S_{Ref}} \int_0^{\frac{b}{2}} Y_{LE}(y) \times C(y) dy \tag{9}$$

### 3.1.2. Fuselage

A fuselage is modeled as a slender circular cylinder of length $L_{FUS}$ and a diameter $D_{FUS}$. The aerodynamic module considers the fuselage to be set on the symmetry plane but with no consideration about its relative position to the wing. Therefore, similar performance will be evaluated by the module for the different configurations presented in Figure 6. The wing dihedral angle is not taken into account.

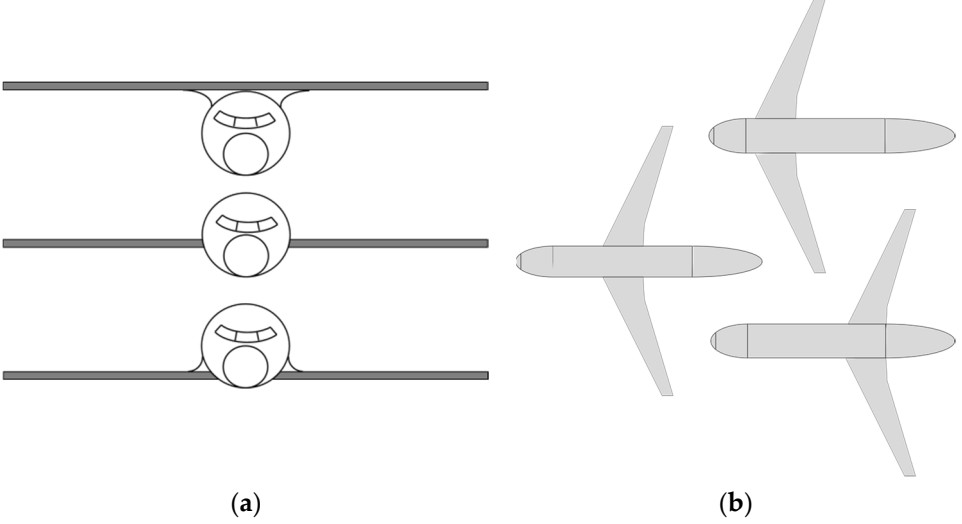

**Figure 6.** Fuselage arrangements with similar aerodynamic performance. (**a**) Front view. (**b**) Top view.

### 3.1.3. Nacelles

The effect of airframe installation on performance is considered for the drag contribution only (there are no propulsion effects considered). The evaluation is therefore similar to a "through-flow nacelle" consideration. Nacelle elements are considered as cylinders. Single flux (NAC1 elements only) or double flux nacelles (NAC1 and NAC2 elements) can be considered (Figure 7), but it is supposed that all the nacelles are identical in the present version of the module.

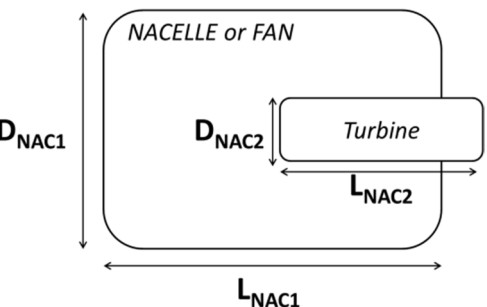

**Figure 7.** Nacelle types considered in the module.

### 3.2. Formulation for Aircraft Lift

### 3.2.1. Estimation of Wing Maximum Lift

By construction, the aerodynamic module considers an ideal elliptic span loading, $KZ(Y)$, for the reference wing, which is fully defined by the load at the symmetry plane $KZ_0$:

$$KZ(Y) = KZ_0 \sqrt{1 - \left(\frac{Y}{\frac{b}{2}}\right)^2} \tag{10}$$

where:

$$KZ_0 = \frac{2S_{WING}}{\pi\left(\frac{b}{2}\right)} \times C_L \tag{11}$$

For a given Y section, the local airfoil lift coefficient corresponds to:

$$C_l(Y) = \frac{KZ(Y)}{C(Y)} \tag{12}$$

Wing maximum lift is considered to be reached when $C_l(Y) = C_{l\max}(Y)$ at one Y section.

### 3.2.2. Lift Slope

The aircraft $\frac{\partial C_L}{\partial \alpha}$ slope ($\alpha$ in rad) considered is the Polhamus formulation [8], with the effects of fuselage taken into account [9]:

$$\frac{\partial C_L}{\partial \alpha} = \frac{\pi\lambda\left[1.07\left[1 + \frac{D_{FUS}}{b}\right]^2\right]}{1 + \sqrt{1 + \lambda^2\left(\frac{1 + \tan\varphi_{50}{}^2 - M^2}{4}\right)}}\left(1 - \frac{D_{FUS}}{b}\right) \tag{13}$$

where $\varphi_{50}$ is the mean wing sweep angle at mid-chord. For FW or BWB configurations, $D_{FUS}$ is set to zero.

Some modules of the MDAO process require the $C_L(\alpha)$ curve as the input, which means that it is necessary to know the zero-lift incidence of the aircraft $\alpha_0$. The aerodynamic module is not able to determine this parameter, but it can be estimated based on preliminary validation exercises on similar configurations. Otherwise, a value of $\alpha_0 = 0°$ can be used.

### 3.3. Formulations for Aircraft Drag Evaluation

Among the different existing drag formulations, the one retained for the aerodynamic module is derived from the one presented by Gur [10]:

$$C_{D_{Total}} = C_{D_{Induced}} + C_{D_{Friction}} + C_{D_{Add}} + C_{D_{Wave}} + C_{D_{Parasitic}} \tag{14}$$

The following sections describe the formulations used for these different terms.

### 3.3.1. Lift-Induced Drag

Only the wing contribution to lift-induced drag is considered with the influence of the fuselage or winglet. The contribution of the tail surfaces to the total lift-induced drag is not considered in the module. The standard formulation is considered for the lift-induced drag coefficient of the reference wing:

$$C_{D_{Induced}} = \frac{C_L^2}{\pi \, \lambda \, Osw} \tag{15}$$

where $\lambda$ is the wing geometric aspect ratio defined in Equation (5), and *Osw* is the Oswald factor that characterizes the deviation from the ideal elliptic span loading. For a given wing planform, several methods are available for the determination of the Oswald factor (see [11] for instance). Most of them consider an Oswald factor such as:

$$Osw = \frac{1}{1 + \delta} \tag{16}$$

where $\delta$ is a parameter based on wing geometrical characteristics. The one proposed in the aerodynamic module is based on the formulation used by Anderson [11,12]:

$$\delta_{Anderson} = \left[ 0.0015 + 0.016(\varepsilon - 0.4)^2 \right] \times \left[ \left( \lambda \sqrt{1 - M^2} \right) - 4.5 \right] \tag{17}$$

leading to:

$$Osw_{Anderson} = \frac{1}{1 + \delta_{Anderson}} \tag{18}$$

However, this formulation is valid for unswept wings only, which is obviously not suitable for BWB configurations. To take the wing sweep angle into account, Hörner [12] proposes a simple correction:

$$Osw_\varphi = Osw_{(\varphi=0)} \times \cos \varphi \tag{19}$$

Some validations carried out on several CFD results analyzed by a far-field drag decomposition tool have shown that the lift-induced drag component computed for swept wings was surrounded by the evaluations made considering $Osw_{Anderson}$ or $Osw_\varphi$ and that a mean value between these two formulations leads to an excellent agreement (Figure 8). Therefore, the formulation retained in the code is simply:

$$Osw_{Wing} = \frac{1 + \cos \varphi}{2} \, Osw_{Anderson} \tag{20}$$

When a fuselage or winglets are considered, the Oswald factor is modified, as described by Nita [11]:

$$Osw_{Wing+Fuselage+WLT} = Osw_{Wing} \times K_{Fus} \times K_{WLT} \tag{21}$$

The correction factor for the fuselage is:

$$K_{Fus} = \left( 1 - 2 \left( \frac{D_{FUS}}{b} \right)^2 \right) \tag{22}$$

$D_{Fus}$ is the fuselage diameter, and b is the total wing span.
The correction factor for the winglets is:

$$K_{WLT} = \frac{1}{Coef_{WLT}} \left( 1 + \frac{2 \, H_{WLT}}{b} \right)^2 \tag{23}$$

where $Coef_{WLT}$ is a correction factor derived from works from Bourdin [13] or Delavenne [14] to take into account the effect of the winglet cant angle on lift-induced drag:

$$Coef_{WLT} = 1.0 + 4.10^{-4}\delta_{WLT} + 1.10^{-5}\delta_{WLT}^2 - 3.10^{-8}\delta_{WLT}^3 - 5.10^{-10}\delta_{WLT}^4 \tag{24}$$

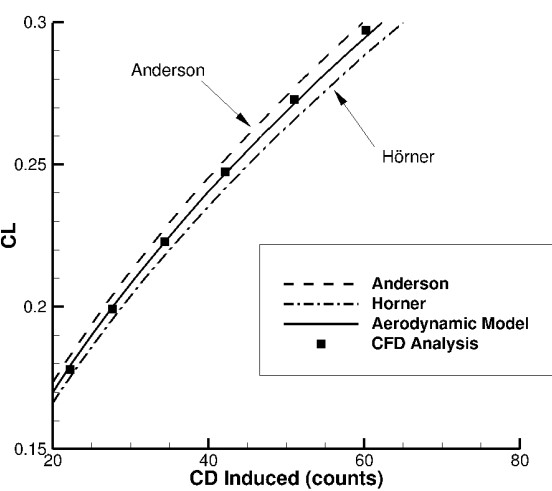

**Figure 8.** Lift-induced drag formulation for swept wings.

$H_{WLT}$ is the winglet height, b is the total reference wing span without winglets, and $\delta_{WLT}$ is the winglet cant angle (in °) (see definitions in Figure 9).

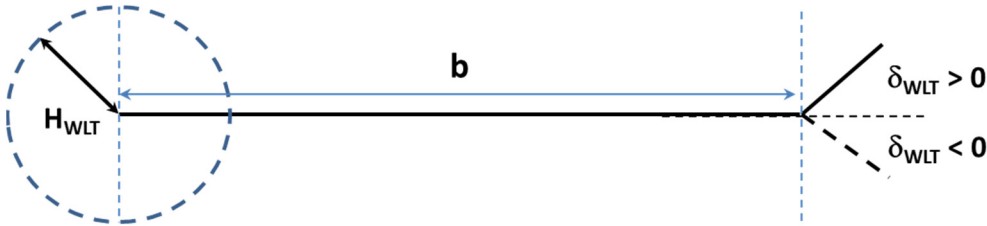

**Figure 9.** Definitions of the winglet parameters.

3.3.2. Friction/Form Drag

According to the methodology described in [9,10], the skin friction and pressure drag of the different components are calculated using the following relation:

$$C_{D_{Friction}} = C_F \cdot FF \cdot \frac{S_{WET}}{S_{REF}} \tag{25}$$

where $C_F$ is a flat plate skin friction coefficient, FF is the form factor of the component, and $S_{WET}$ is the wetted area of this component.

- **Friction coefficient $C_F$**

For a flat plate of length L, the friction coefficient is obtained considering the definition of the boundary layer momentum thickness. At the end of the plate, the integrated friction coefficient is:

$$C_F = 2\frac{\Theta_L}{C(Y)} \tag{26}$$

If the flow is laminar, the use of the Blasius relationships leads to:

$$C_{F_{Lam}} = \frac{1.328}{\sqrt{Re_L}} \tag{27}$$

For a turbulent flow, the friction coefficient is obtained by the compressible Schlichting relation [9]:

$$C_{F_{Turb}} = \frac{0.455}{(log_{10}Re_L)^{2.58}(1 + 0.144M^2)^{0.65}} \tag{28}$$

For wings considering natural laminar flow (NLF) or hybrid laminar flow (HLF) technologies, it is possible to have an estimation of the maximum laminar extent $X_{Lam_{max}}$ for the wing sections as a function of the local Reynolds number and the $\varphi_{25}$ sweep angle (Figure 10, from [15]). This maximum laminar extent occurs for a design lift coefficient ($C_{L_{Adapt}}$), and the change in transition location due to the modification of the pressure gradients over the airfoil when the angle of attack changes has to be taken into account.

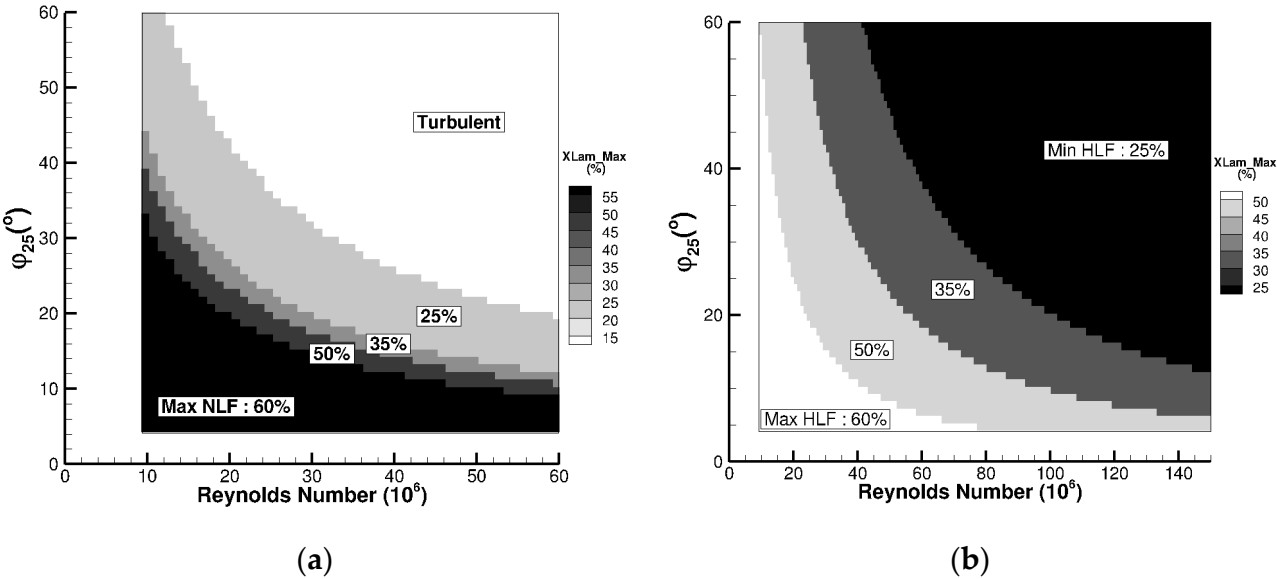

**Figure 10.** Maximum possible laminar flow extent as a function of Reynolds number and wing sweep angle. (**a**) NLF. (**b**) HLF.

Though the module does not consider any airfoil shape, it is therefore necessary to consider a "suction" and a "pressure" side for the wing surfaces in order to take these changes in transition location into account. The module uses the simplified model presented in Figure 11.

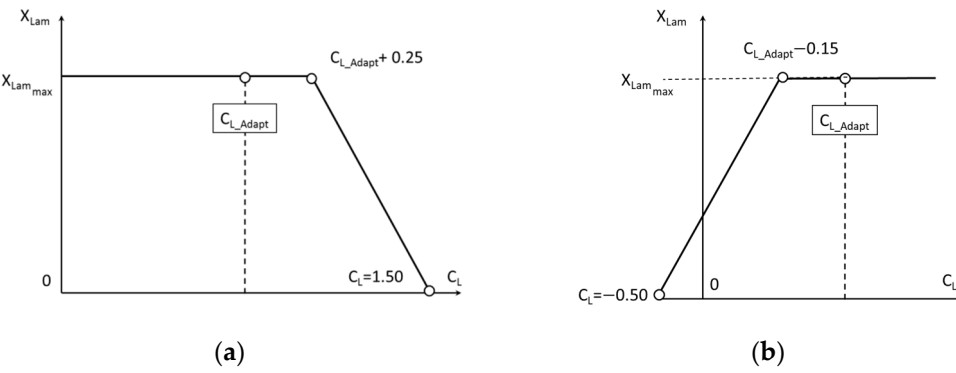

**Figure 11.** Simplified model used for transition location on wing surfaces. (**a**) Model for transition change on wing suction side. (**b**) Model for transition change on wing pressure side.

Once the portion of laminar flow ($X_{Lam}$) is estimated for the considered wing section, the global friction coefficient is obtained via the computation of the boundary layer momentum thickness at the trailing-edge through the use of a "fictitious turbulent length" ($L_{Fict}$ in Figure 12).

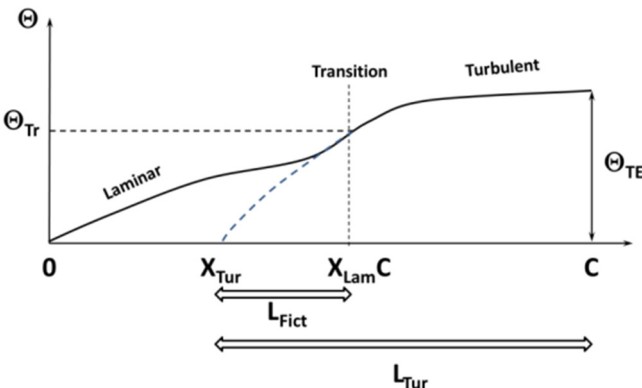

**Figure 12.** Estimation of $\Theta$ at airfoil trailing-edge for transitional flow.

Firstly, the momentum thickness at the transition point ($\Theta_{Tr}$) is computed using the Blasius relationships.

$$\Theta_{Tr} = (X_{Lam} \times C(Y)) \frac{0.664}{\sqrt{Re_{(L=X_{Lam} \times C(Y))}}} \tag{29}$$

Then, using the Michel relationships [16], the length $L_{Fict}$ of a fully turbulent boundary layer with the same $\Theta_{Tr}$ momentum thickness is estimated as:

$$L_{Fict} = \left[ \frac{1}{0.02208} \Theta_{Tr} \left( \frac{Re_{(L=X_{Lam} \times C(Y))}}{X_{Lam} \times C(Y)} \right)^{1/6} \right]^{1.2} \tag{30}$$

Finally, $\Theta_{TE}$ is computed as the momentum thickness of a turbulent flow that develops over a length of $L_{Tur}$:

$$L_{Tur} = L_{Fict} + (1 - X_{Lam}) \times C(Y) \tag{31}$$

$$\Theta_{TE} = 0.02208 \frac{L_{Tur}}{\left( Re_{L_{Tur}} \right)^{1/6}} \tag{32}$$

The friction coefficient for transitional flow on one surface is finally obtained by introducing $\Theta_{TE}$ from Equation (32) into Equation (26).

- **Form Factors FF**

There are several models in the literature available for form factors (see for instance [10] for a comparison of different formulations for $FF_{WING}$ or for body of revolutions). Table 2 presents the ones retained in the aerodynamic module and the corresponding $S_{WET}$.

**Table 2.** Form factors and wetted area considered in the aerodynamic module.

| Element | Form Factor | $S_{Wet}$ |
|---|---|---|
| Wing [17] | $FF_{WING} = \left[ 3.4004 \left( \frac{t}{c} \right) - 0.4578 \left( \frac{t}{c} \right)^2 + 13.0119 \left( \frac{t}{c} \right)^3 \right] cos^2 \varphi_{50} + 1$ | $2\, S_{WING}$ |
| Fuselage [9] | $FF_{FUS} = 1 + \frac{60}{\left( \frac{L_{FUS}}{D_{FUS}} \right)^3} + 0.0025 \left( \frac{L_{FUS}}{D_{FUS}} \right)$ | $\pi\, L_{FUS}\, D_{FUS}$ |
| Winglets [17] | $FF_{WLT} = 1 + 3.52 \left( \frac{t}{c} \right)_{WLT} cos\, \varphi_{WLT}$ | $2\, S_{WLT}$ |
| Tail surfaces [17] | $FF_{TAIL} = 1 + 3.52 \left( \frac{t}{c} \right)_{TAIL} cos\, \varphi_{TAIL}$ | $2\, S_{TAIL}$ |
| Nacelles [18] | $FF_{NAC} = 1 + 0.35 \left( \frac{D_{NAC}}{L_{NAC}} \right)$ | $2\pi\, D_{NAC}\, L_{NAC}$ |

### 3.3.3. Interaction Effects for Nacelles

In addition to the simple geometrical effect on the friction drag, some interaction effects of the nacelles with a surface (wing or fuselage) are considered through the distance parameter $Z_{NAC}$ (Figure 13). The positive values of $Z_{NAC}$ are for standard nacelle arrangements. The negative values of $Z_{NAC}$ are for buried nacelles. On the basis of these geometrical characteristics, the interaction coefficient $Q_N$ is defined according to the following statement given in [9]: *"For a nacelle or external store mounted directly on the fuselage or wing, the interference factor $Q_N$ is about 1.5. If the nacelle or store is mounted less than about one diameter away, the $Q_N$ factor is about 1.3. If it is mounted much beyond one diameter, the $Q_N$ factor approaches 1.0."*

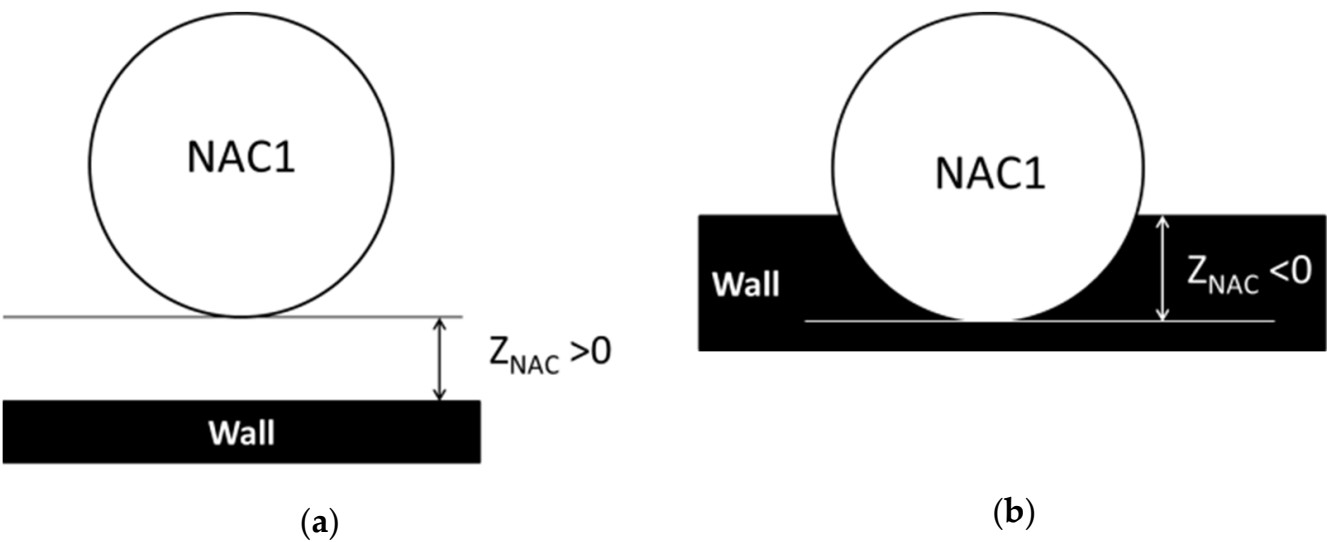

**Figure 13.** Definition of the $Z_{NAC}$ parameter. (**a**) "Standard". (**b**) "Buried".

For the positive values of $Z_{NAC}$, a linear evolution of $Q_N$ is considered in the module:

$$\text{If } Z_{NAC} > 0; \ Q_N = max(1, 1.5 - 0.25 \ \frac{Z_{NAC}}{D_{Fan}}) \tag{33}$$

For the negative values of $Z_{NAC}$, the interference factor is considered at its maximum value (i.e., 1.5), but we take into account the modification of the wetted surface of the nacelle by imposing a minimum value of 1.0 to the factor. It leads to the following relationship:

$$\text{If } Z_{NAC} < 0; \ Q_N = max\left(1.0, \ 1.5 \times \left(1 - \frac{acos\left(1 + 2\frac{Z_{NAC}}{D_{Fan}}\right)}{\pi}\right)\right) \tag{34}$$

Once this coefficient is evaluated, the global friction drag coefficient for the $N_{ENG}$ nacelles is obtained by the following relation:

$$C_{D_{ENG}} = N_{ENG} \times \left(Q_N C_{D_{NAC1}} + C_{D_{NAC2}}\right) \tag{35}$$

where the individual friction drag coefficients for NAC1 and NAC2 are computed according to Equation (25) with the form factor and wetted areas given in Table 2.

### 3.4. Additional Drag Due to Lift

An additional profile of drag due to lift is considered for the wing. For each wing segment, the additive drag is estimated according to the relation from [18]:

$$C_{D_{ADD}} = 0.75 \, C_{D_{ADD_{Ref}}} \left( \frac{C_L - C_{L0}}{C_{L_{Max}} - C_{L0}} \right)^2 \sqrt{1 - (M cos \varphi_{25})^2} \, \frac{S_{Segment}}{S_{REF}} \tag{36}$$

where:

$$C_{D_{ADD_{Ref}}} = \left[ 0.010 \, C_{L_{Max}} - 0.0046 \left( 1 + 2.75 \left( \frac{t}{c} \right) + 100 \left( \frac{t}{c} \right)^4 \right) \right] cos^3 \varphi_{25} \tag{37}$$

$C_{L_{Max}}$ is the maximum lift coefficient of the wing estimated from Equation (12), and $C_{L0}$ is the minimum drag lift coefficient provided in the data file.

### 3.5. Wave Drag

For aircraft missions at transonic flight conditions, the drag increase due to compressibility effects has to be considered. In the module, the estimation of this drag component is based on the Korn equation [10]. In the original work, Korn gives an estimation of the divergence Mach number for an airfoil as:

$$M_{DD} = K_A - \frac{1}{10} C_l - \left( \frac{t}{c} \right) \tag{38}$$

The $K_A$ coefficient, referred to as the "Korn factor", is an airfoil technology coefficient, depending on the nature of the airfoil. It is proposed to use $K_A = 0.95$ for "modern" supercritical airfoils and $K_A = 0.87$ for "conventional" airfoils. For wing segments corresponding to a transition area with the fuselage on a BWB configuration, $K_A = 0.90$ can be used. However, Equation (38) is valid for 2D airfoils only. In order to have an estimation of $M_{DD}$ for a wing section, the simple swept wing theory, considering a normal to the leading-edge flow assumption, can be used:

$$M_{2D} = M_{3D} \cos \varphi \tag{39}$$

$$C_l = C_{L_{3D}} \frac{1}{cos^2 \varphi} \tag{40}$$

$$\left( \frac{t}{c} \right)_{2D} = \left( \frac{t}{c} \right)_{3D} \frac{1}{\cos \varphi} \tag{41}$$

Introducing Equations (39)–(41) into Equation (38), the following relation for the 3D swept wing section is obtained:

$$M_{DD} = \frac{K_A}{\cos \varphi} - \frac{1}{10} \frac{C_l}{cos^3 \varphi} - \frac{1}{cos^2 \varphi} \left( \frac{t}{c} \right) \tag{42}$$

Then, the local critical Mach number $M_{Cr}$ is estimated according to:

$$M_{Cr} = M_{DD} - \left( \frac{0.1}{80} \right)^{\frac{1}{3}} = M_{DD} - 0.108 \tag{43}$$

Finally, for a given wing subsection, the wave drag contribution is obtained:

$$If \; M > M_{Cr} \; ; \; C_{D \, WAVE} = 20 \, (M - M_{Cr})^4 \, \frac{S_{element}}{S_{REF}} \tag{44}$$

The total wave drag of the wing can be found by adding the contributions of all the NDY wing subsections.

*3.6. Parasitic Drag*

Finally, an additional parasitic drag ($C_{D_{PARASITIC}}$) (due to protuberances, antenna, probes, paint, etc.) is considered. A general formulation used in the pre-design phase is to consider this contribution as a percentage of the total friction drag:

$$C_{D_{PARASITIC}} = X_{PARA} \times C_{D_{FRICTION}} \tag{45}$$

For an aircraft performance estimation, $X_{PARA}$ = 0.025 is generally used. However, for a validation purpose with comparisons with the CFD results or experimental data in a wind tunnel, $X_{PARA}$ = 0.0 has to be used.

**4. Module Validation**

The following chapters present the validation of the aerodynamic model on standard tube and wing or blended wing–body configurations. The results obtained by the module are compared to the CFD results or experimental data in terms of overall performance and drag components when possible.

Note that all the drag coefficients presented are expressed in drag counts.

*4.1. Standard Tube and Wing Configurations*

4.1.1. ONERA NOVA Configuration

The first configuration considered is the Nextgen ONERA Versatile Aircraft (NOVA) [19] (Figure 14), which is designed with a downward winglet element and different engine configurations.

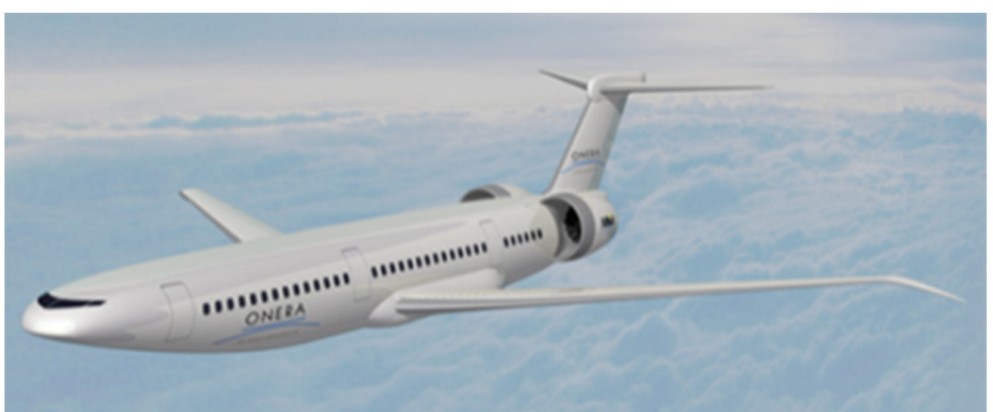

**Figure 14.** ONERA NOVA Aircraft.

The aerodynamic flow conditions for this test case are a Mach number of M = 0.82 at an altitude of Z = 11,300 m. For this configuration, drag decompositions using the *ffd72* tools [20] of a different configuration are available, which makes it possible to compare the lift-induced drag for the clean wing and the wing equipped with the winglet with a negative cant angle $\delta_{WLT} = -18°$.

The comparison with the prediction from the aerodynamic module (Figure 15) shows a quite good prediction of this drag component, even though only four wing segments are used for the wing definition.

NOVA is a platform used to investigate the effect of the integration of ultra-high bypass ratio (UHBR) engines. Different engine arrangements are available (Figure 16) that make possible a first validation of the different nacelle interference ratios on drag.

Table 3 compares the drag increase for different engine installations for the NOVA aircraft between the CFD results and the estimation from the module. A quite good agreement can be found for the different cases proving that the formulation of the interference factor for the nacelle is realistic.

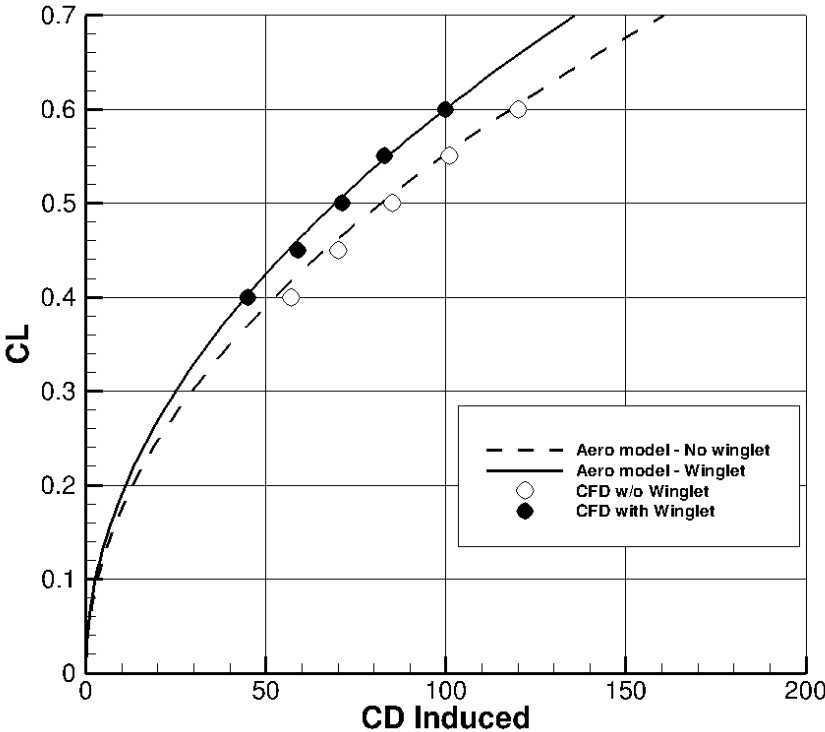

**Figure 15.** NOVA configuration—winglet effect ($\delta_{WLT}$ = −18°). Lines—aerodynamic module; symbols—CFD.

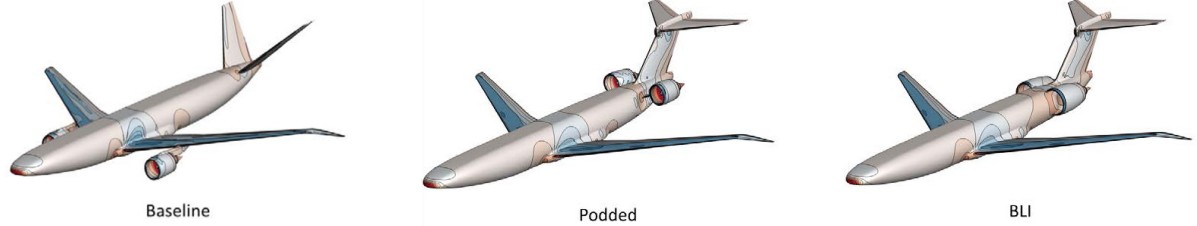

**Figure 16.** NOVA configurations used for UHBR engine integration.

**Table 3.** NOVA configuration—drag increase (drag counts) for different engine installations.

|  | $Z_{NAC}$ (m) | Aerodynamic Module | | CFD |
|---|---|---|---|---|
| $C_L$ = 0.50 | | $C_D$ | $\Delta C_D$ | $\Delta C_D$ |
| Ref. (No Engine) | − | 210 | − | − |
| Baseline | +10.00 | 232 | +22 | +28 |
| Podded | +0.75 | 244 | +34 | +34 |
| BLI | −0.50 | 226 | +16 | +18 |

### 4.1.2. NASA Common Research Model (CRM)

The second configuration considered for the validation exercise is the NASA CRM model (Figure 17), for which both experimental and numerical data are available [21–23].

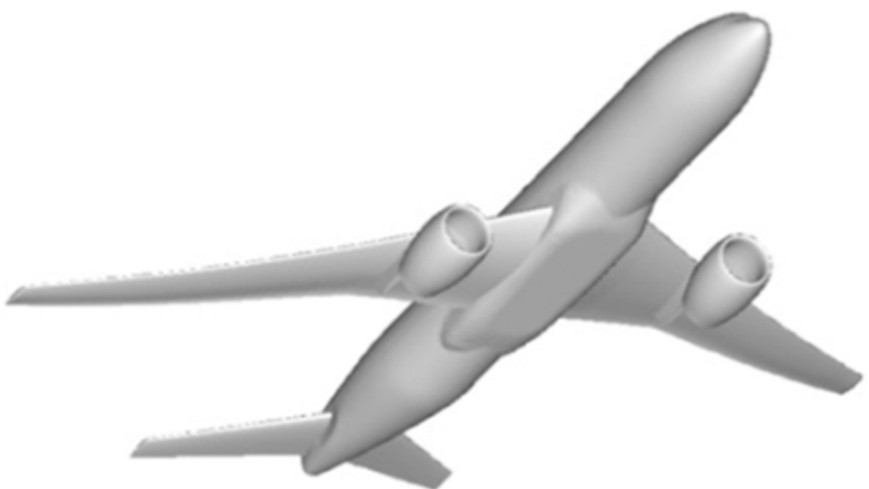

**Figure 17.** Common research model (CRM) reference configuration.

For the evaluation of the performance, the module considered only six wing sections (Figure 18).

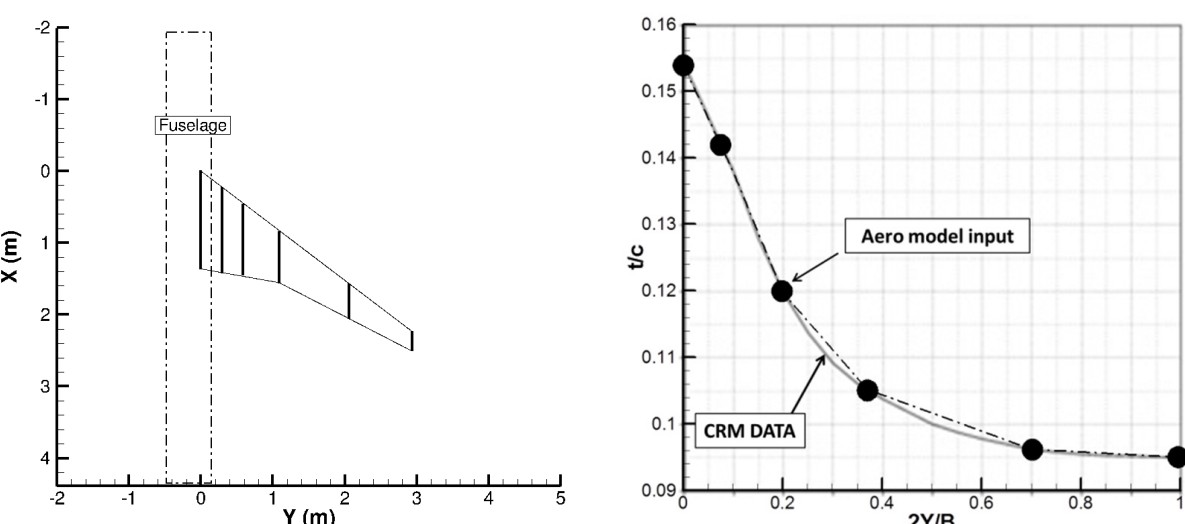

**Figure 18.** CRM wing—data used by the aerodynamic module.

The flow conditions considered are the ones used for the different AIAA Drag Prediction Workshop (DPW) exercises: model scale, 1/10; Mach number = 0.850; and Reynolds number = $5.36 \times 10^6$.

Note that for the CRM configuration, the reference area used for the force coefficients is not the geometrical wing one, as considered by the module ($S_{Wing}$ = 411.806 m$^2$), but a modified trapezoidal one ("Wimpress" area = 383.69 m$^2$). For the present comparisons, the force coefficients considered for the CRM database (from CFD or experiments) are therefore corrected in order to deal with this difference in $S_{REF}$ with a factor of 0.93172.

The first validation exercise considered the effect of the fuselage on the $C_L(\alpha)$ curve. Figure 19 compares the $C_L(\alpha)$ curves from the aerodynamic module, with or without a fuselage taken into account, with the CFD results from ONERA obtained within the DPW framework. The module considers a zero-lift incidence of $\alpha_0 = -1.55°$ to improve the comparison. A quite good agreement can be observed for the estimation of the lift slope of the wing–fuselage arrangement by the module with the CFD data.

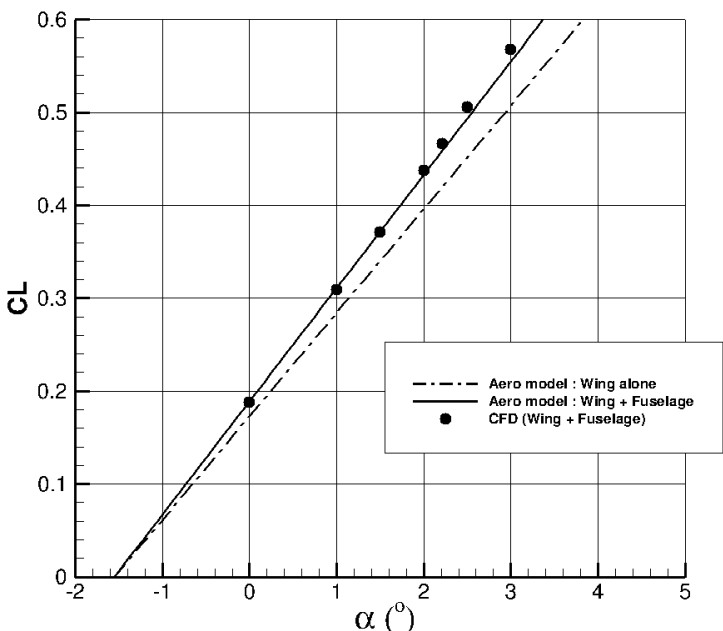

**Figure 19.** CRM configuration—fuselage effect on the $C_L(\alpha)$ curve. Lines—aerodynamic module; symbols—CFD.

Regarding the overall drag estimation, Figure 20 compares the $C_L(C_D)$ curves computed by the module with the CFD results from ONERA [23]. The drag coefficients from the CFD results were obtained by the *ffd72* far-field analysis tool [20] that allows the elimination of the artificial spurious drag from the numerical solution and gives the drag breakdown of the physical components (lift-induced, viscous and wave). It can be seen that the agreement is quite good ($\Delta C_D = 1$ d.c. maximum) between the module evaluation and the CFD results up to lift coefficients of around 0.50 (using $S_{REF} = S_{WING}$) as well as for the wing–fuselage configuration and for the wing–fuselage–horizontal tail planes case.

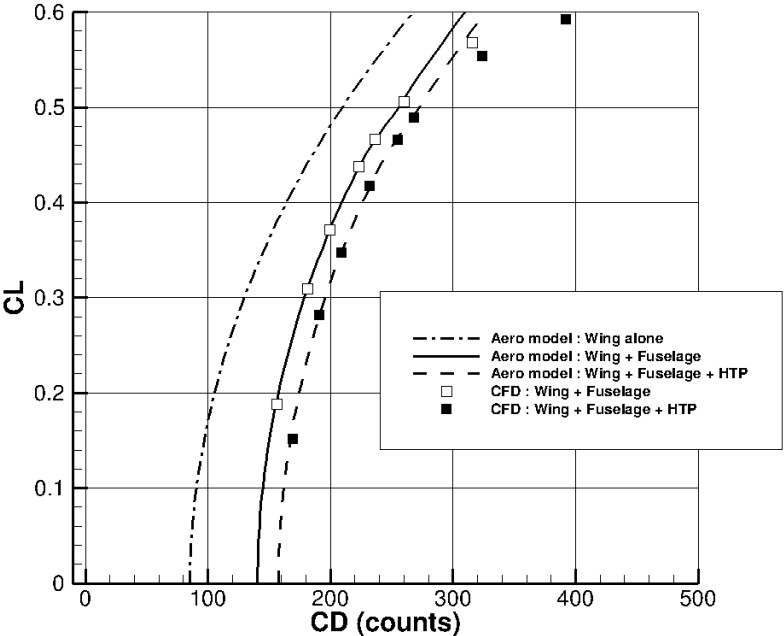

**Figure 20.** CRM configuration—$C_L(C_D)$ curves ($M = 0.85$, $Re = 5 \times 10^6$). Lines—aerodynamic module; symbols—CFD.

When considering the different drag components, namely, lift-induced, viscous and wave drag, a very good agreement can also be found (Figure 21). Some discrepancies can be observed for $C_L$ values higher than 0.50 for viscous and wave drag, but in these conditions, a separation can be found on the wing by CFD, which is not considered in the aerodynamic module formulations.

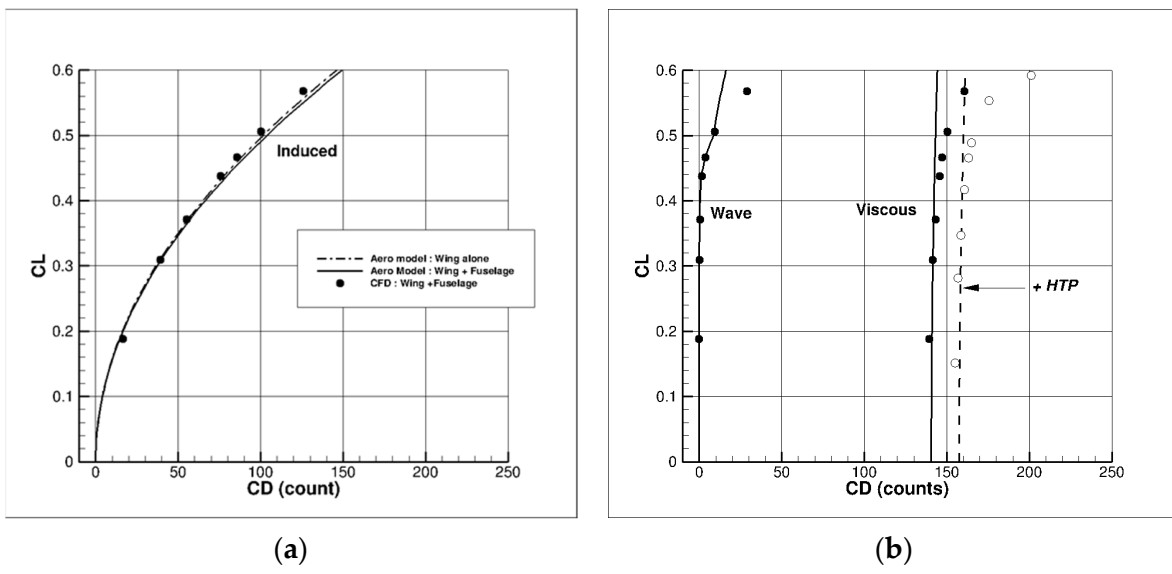

(**a**)  (**b**)

**Figure 21.** CRM configuration—drag components. Lines—aerodynamic module; symbols—CFD. (**a**) Lift-induced. (**b**) Wave and viscous.

Finally, some experimental data are provided on the CRM website for a wing equipped with or without through-flow nacelles. Due to their relative position with the wing surface, nacelles are considered as out of interaction ($Q_N = 1$). It can be seen that the drag increment found experimentally is correctly predicted by the aerodynamic module (Figure 22).

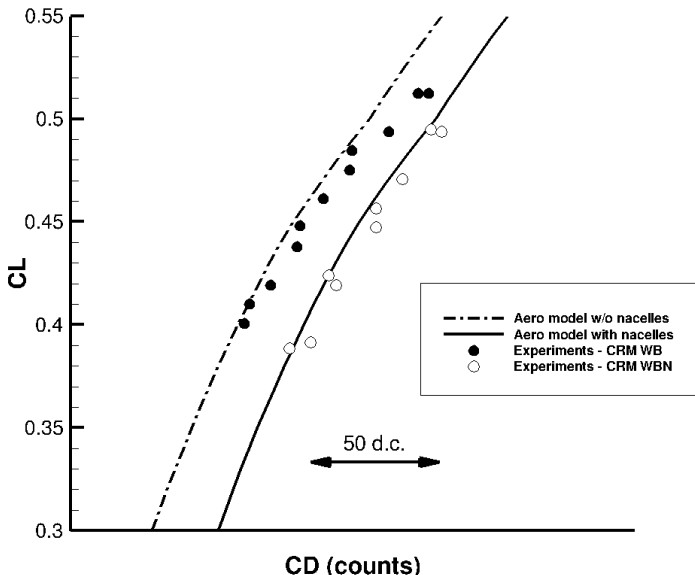

**Figure 22.** CRM configuration—Nacelle (TFN) effect. Lines—aerodynamic module; symbols—experiments.

### 4.1.3. CRM-NLF Configuration

The next validation exercise considers the CRM model with a modified wing with a natural laminar flow on the upper surface. This CRM-NLF wing has been designed for tests in the NASA NTF wind tunnel [24–26], and some experimental results are provided on

the NASA CRM website. The data used for the CRM reference plane are modified for the inboard wing section in order to take the wing planform change (Figure 23) into account. Similar to the CRM reference case, the experimental data are corrected in order to take the change in the reference wing area into account.

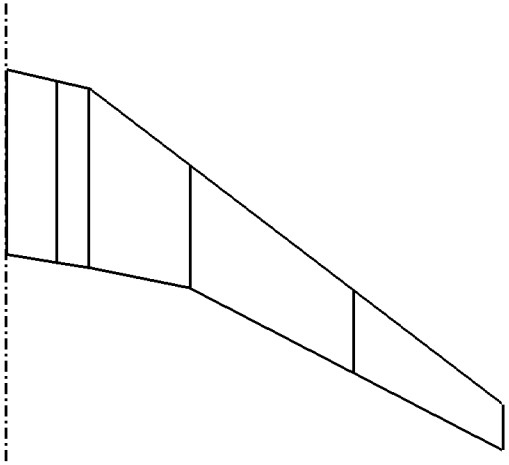

**Figure 23.** CRM-NLF geometry used by the aerodynamic module.

The flow conditions used for the validation are similar to the NTF ones: a Mach number of M = 0.8565 and a Reynolds number of $15 \times 10^6$. The NLF capabilities of the aerodynamic module are considered for this validation exercise, but wing lower surfaces are considered turbulent, as during the wind tunnel tests. The estimated maximum location of the transition point on different wing sections is presented in Figure 24 at $C_L = 0.50$ for the wind tunnel flow conditions. It can be seen that an extended natural laminar flow is possible (around 50%) for a large portion of the wing, except for some portions of the inboard wing, which are more limited in terms of NLF capabilities.

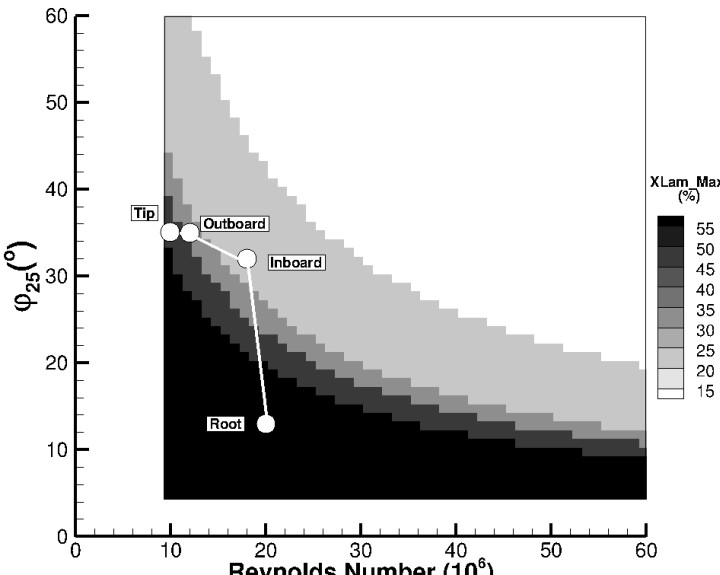

**Figure 24.** CRM-NLF configuration—estimated maximum transition location by the module (M = 0.8565, Re = 15 × 10^6, $C_L$ = 0.50).

Then, these values are considered as the $X_{Lam_{max}}$ parameter, as shown in Figure 11, the location of the transition point $X_{Lam}$ is computed for each $C_L$ considering $C_{L_{Adapt}} = 0.50$. The comparison of the transition line estimated by the module at $C_L = 0.425$, with infrared pictures available on the CRM website, is presented in Figure 25. It can be seen that, though

not exact, the laminar flow extent estimated by the module is quite realistic, especially on the outboard wing.

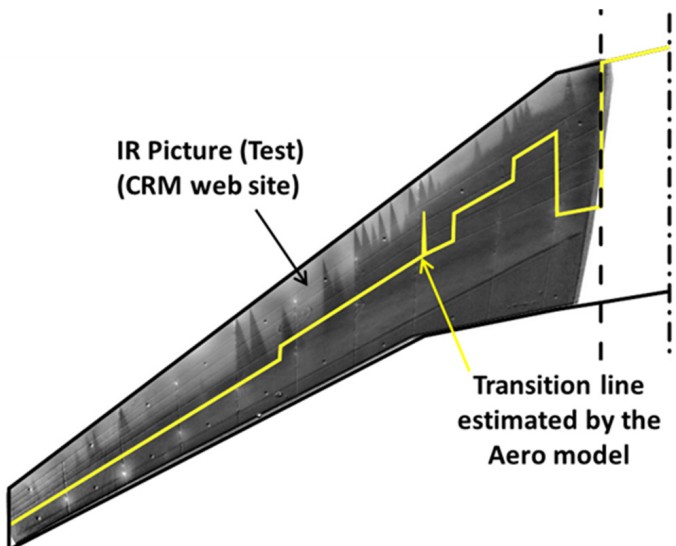

**Figure 25.** CRM-NLF configuration—transition line estimated by the module at $C_L = 0.425$—comparison with experimental infrared image.

Moreover, of most importance is the evaluation of the effect of this laminar flow extent on the overall performance. Figure 26 compares the $C_L(\alpha)$ and the $C_L(C_D)$ curves from the aerodynamic module with the experimental data. A zero-lift angle of attack of $\alpha_0 = -0.90°$ is considered to improve the comparison for the $C_L(\alpha)$ curve. For comparison, the performances estimated by the module in fully turbulent mode are presented in dashed lines. It can be seen that the agreement on the drag evaluation is quite good for NLF flow conditions around the different $C_L$ values available (between 1.5 d.c. and 3 d.c.).

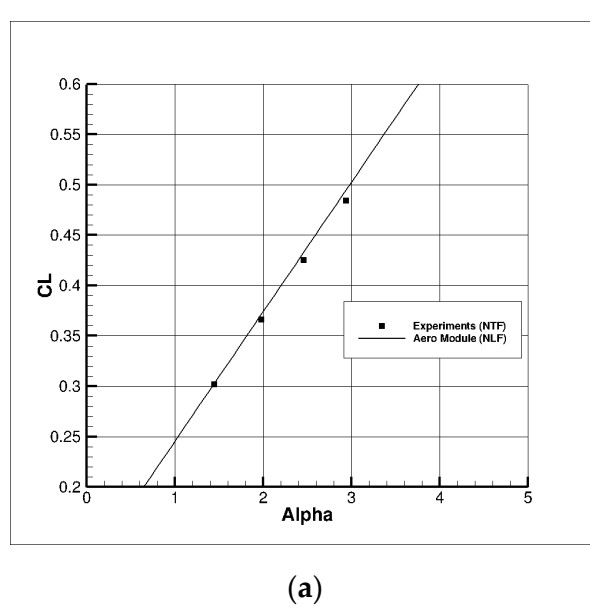

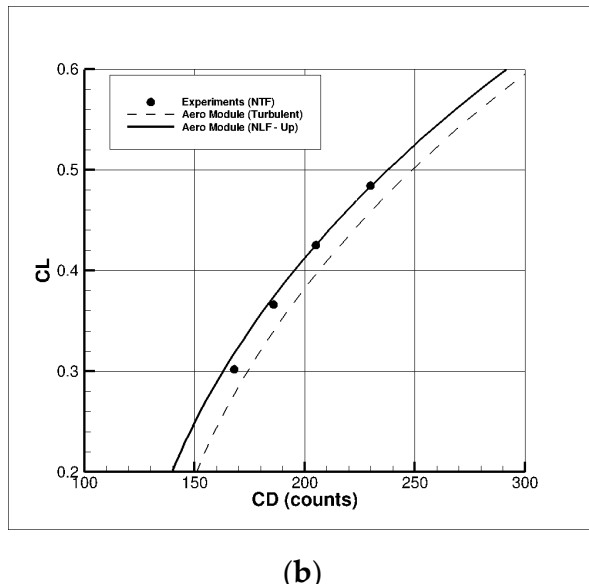

**(a)**

**(b)**

**Figure 26.** CRM-NLF configuration—global aerodynamic performances ($M = 0.8565$, $Re = 15 \times 10^6$). Lines—aerodynamic module; symbols—experiments. (**a**) $C_L(\alpha)$. (**b**) $C_L(C_D)$.

### 4.1.4. NLF Regional Aircraft Configuration

The CRM-NLF configuration considered laminar flow on the upper wing only. In order to evaluate the performance prediction capabilities of the aerodynamic model for a wing in

which laminar flow extents on both surfaces, we considered the AG2-NLF wing design by ONERA in the framework of the Clean Sky 2 AIRGREEN2 program. The reference aircraft considered is a 90-pax turboprop configuration with wing airfoils redesigned by ONERA at cruise conditions for natural laminar flow capabilities [27–29]. This configuration is referred to as AG2-NLF. Figure 27 presents the complete configuration, which considers some winglets, an under-carriage fairing and a wing–body junction Karman that are not modeled in the aerodynamic module. In order to take into account the effect of the Karman on the viscous drag, the first wing section is considered a thick airfoil (t/c = 40%) with a leading-edge sweep of 30° at the symmetry plane.

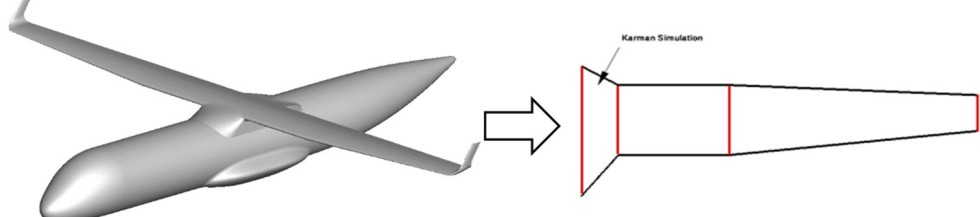

**Figure 27.** AG2-NLF configuration—data used by the aerodynamic module (4 wing sections).

The flow conditions are a Mach number of M = 0.52 at an altitude of Z = 6100 m (20,000 ft). The NLF capabilities of the module are evaluated by comparison with 3D RANS results obtained by the ONERA *elsA* software with integrated transition prediction capabilities [30]. The results are presented in Figure 28. The zero-lift angles of attack considered for the $C_L(\alpha)$ curves are $\alpha_0 = -1.7°$ for turbulent conditions and $\alpha_0 = -1.96°$ for the NLF aircraft. It can be seen that the agreement on the drag evaluation is quite good for both the turbulent and NLF flow conditions at around $C_{L\_Adapt} = 0.50$, with a difference of less than 1 point in LoD.

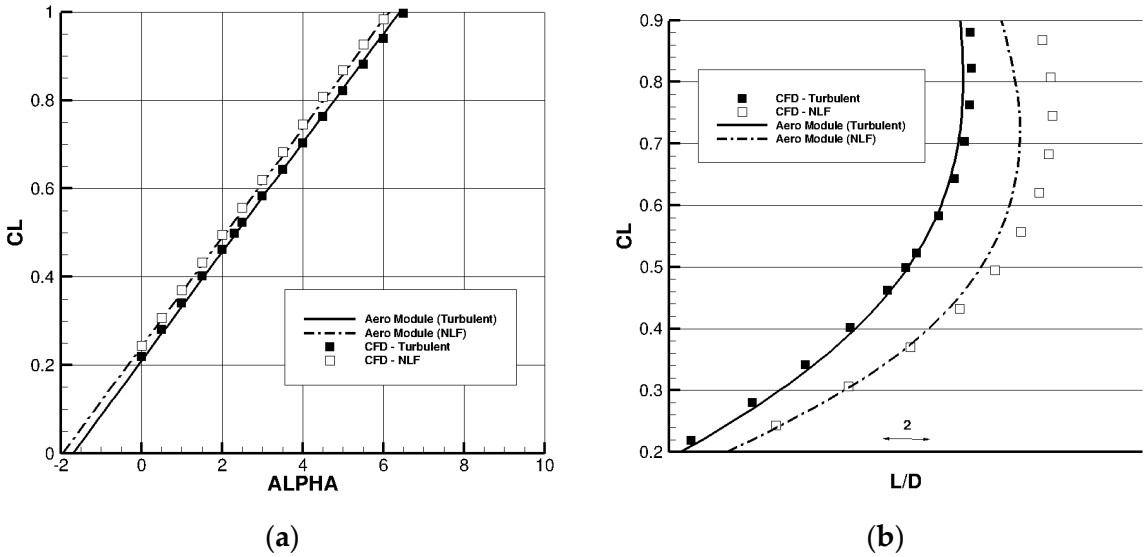

**(a)**        **(b)**

**Figure 28.** AG2-NLF configuration—global aerodynamic performances (M = 0.52, Re = 17.5 × $10^6$). Lines—aerodynamic module; symbols—CFD. (**a**) $C_L(\alpha)$. (**b**) $C_L$(LoD).

### 4.2. BWB Configurations

As the module is used for an MDAO design process of BWB configurations, it is important to check if the formulations used are valid for such configurations with high values of leading-edge sweep angle.

### 4.2.1. AVECA

The first BWB configuration considered is one of the optimized planforms of the AVECA project [31,32]. AVECA is a long-range BWB configuration designed by Airbus. Several wing planforms, considering some changes in the geometrical characteristics or different volume constraints for the passenger cabin or cargo, have been optimized by ONERA using the adjoint method. One of the optimized configurations is selected for the validation purpose considering only four wing segments or five wing sections (Figure 29), which is very crude. The flow conditions are a Mach number of M = 0.850 at an altitude of Z = 11,000 m. The results of the aerodynamic module are compared to the CFD results post-processed with the *ffd72* tool in order to obtain the drag breakdown between the different drag components. Note that the engines were not considered for both CFD and the module.

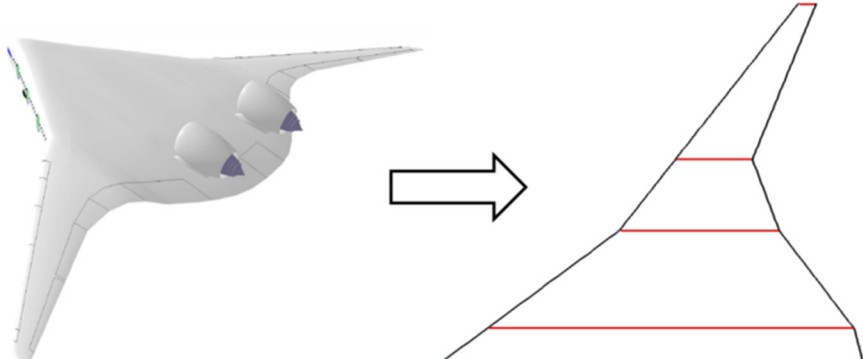

**Figure 29.** AVECA BWB configuration—data used by the aerodynamic module (5 wing sections).

Figure 30 compares the $C_L(\alpha)$, with a zero-lift angle of attack of $\alpha_0 = +0.175°$ to improve the comparison, and the $C_L(C_D)$ curves from the aerodynamic module (continuous lines) with the CFD results (symbols). It can be seen that the agreement on the lift slope and on the total drag evaluation is quite good. Similarly, a very good correlation can be found on the different drag components (lift-induced, viscous and wave) presented in Figure 31, with a small underestimation of the viscous drag by the module.

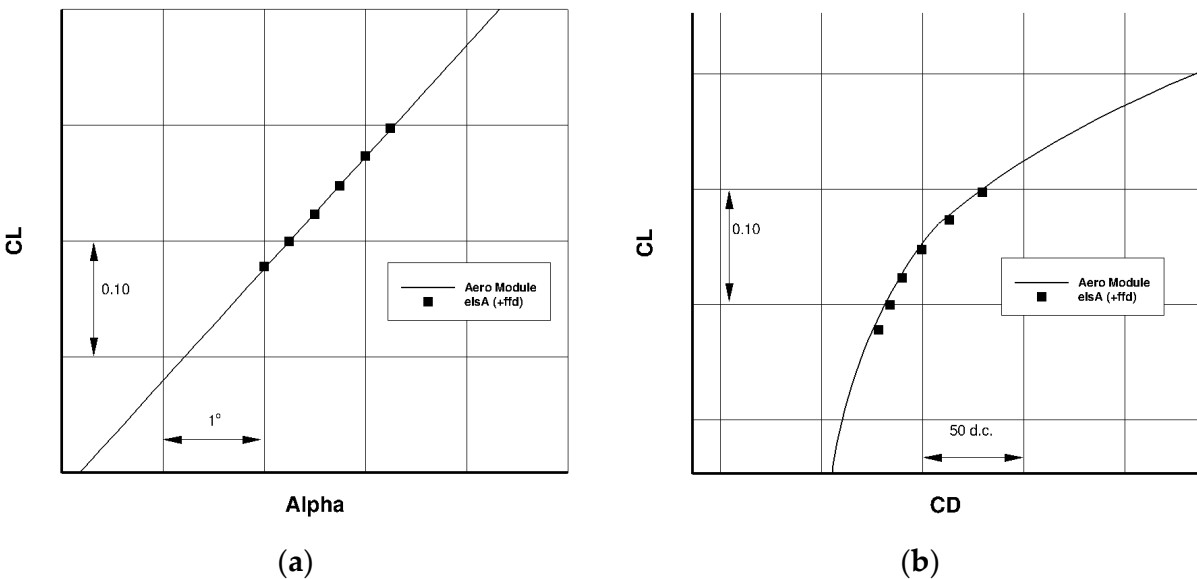

**Figure 30.** AVECA BWB configuration—global aerodynamic performances (M = 0.85, Re = 180 × $10^6$). Lines—aerodynamic module; symbols—CFD. (**a**) $C_L(\alpha)$. (**b**) $C_L(C_D)$.

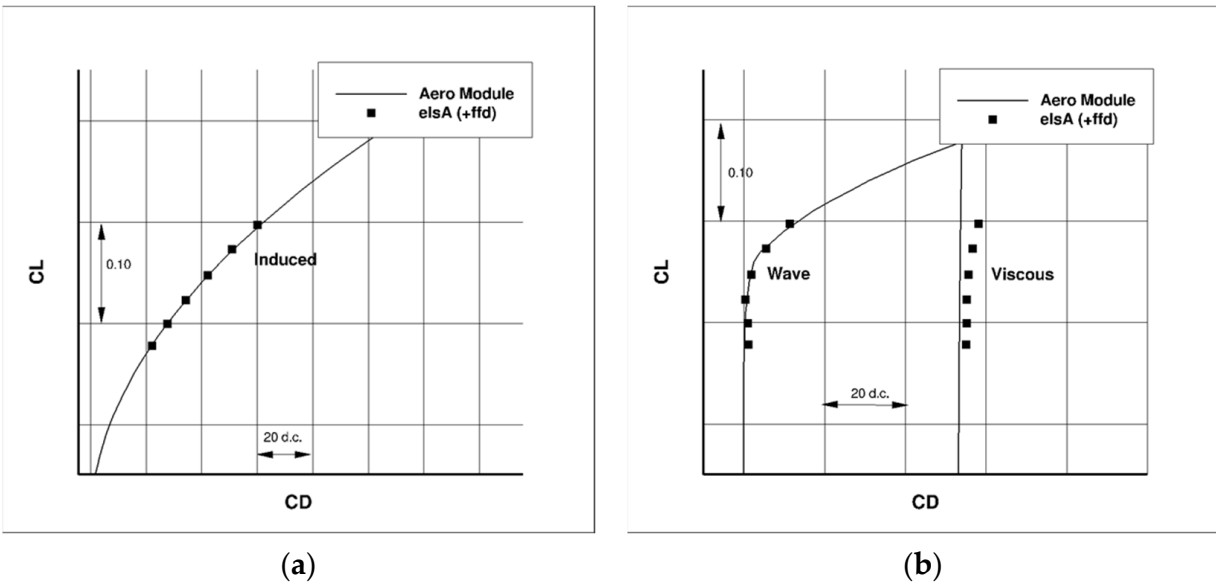

**Figure 31.** AVECA BWB configuration—drag components. Lines—aerodynamic module; symbols—CFD. (**a**) Lift-induced. (**b**) Wave and viscous.

### 4.2.2. NACOR-SMILE Configuration

The second BWB configuration considered for the validation exercise is the optimized SMILE planform from the CS2 NACOR program. This configuration is designed for a short–medium range (SMR) mission, similar to the A320 aircraft. Therefore, compared to the AVECA configuration, its span is much smaller (36 m for SMILE, 80 m for AVECA). It should be noted that the aerodynamic module is used within the OAD-MDAO definition phase, but the final optimization details are carried out using the CFD methods [33].

Figure 32 compares a CAD rendering of the final optimized shape with the data used by the aerodynamic module. Only six wing sections are considered to model the BWB planform. Winglet and nacelles are also considered for the performance evaluations by the module.

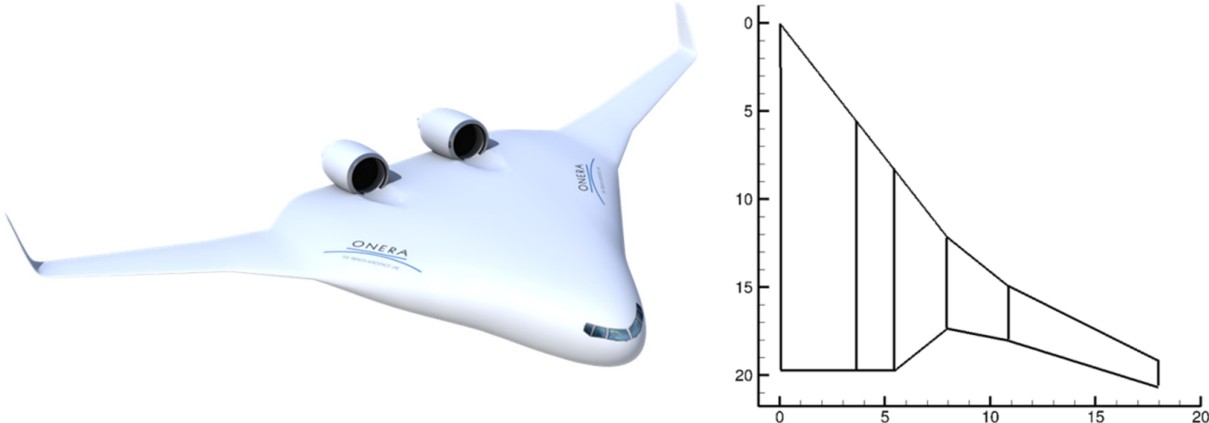

**Figure 32.** SMILE BWB configuration—data used by the aerodynamic module (6 wing sections).

Figure 33 compares the $C_L(\alpha)$, with a zero-lift angle of attack of $\alpha_0 = -0.225°$ to improve the comparison, and the $C_L(C_D)$ curves from the module with the CFD results. It can be seen that the agreement on the lift slope and on the drag evaluation is quite good. Similarly, a very good correlation can be found on the different drag components (lift-induced, viscous and wave) presented in Figure 34.

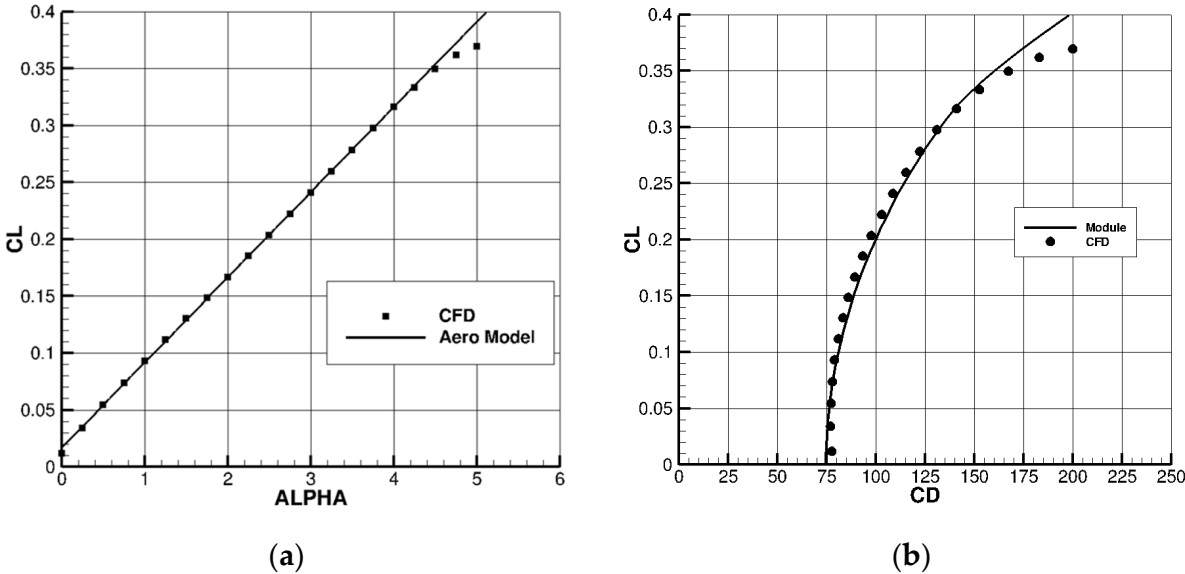

**Figure 33.** SMILE BWB configuration—global aerodynamic performances (M = 0.78, Alt. = 41,000 ft). Lines—aerodynamic module; symbols—CFD. (**a**) $C_L(\alpha)$. (**b**) $C_L(C_D)$.

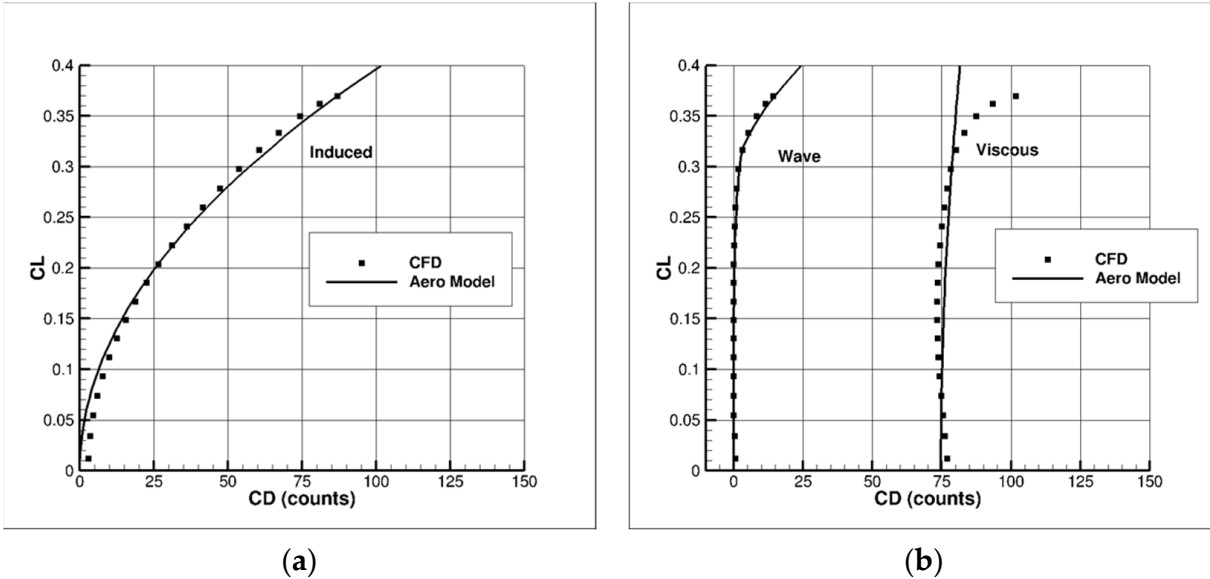

**Figure 34.** SMILE BWB configuration—drag components. Lines—aerodynamic module; symbols—CFD. (**a**) Lift-induced. (**b**) Wave and viscous.

## 5. Computing Performances

The use of analytical formula allows quite fast computing times for this module, which makes it quite useful for its use within a multi-disciplinary optimization/design process where a large number of configurations are considered. As an example, for a complete aircraft configuration, as the CRM case presented, the data file considers about 90 lines only (including comments), and the computing time to obtain one complete $C_L(C_D)$ curve (~120 values) is in the order of 1 sec on a working station (DELL Precision 3630).

## 6. Conclusions

In the framework of a multi-disciplinary design analysis and optimization process for BWB configurations, an analytic aerodynamic module was developed in order to evaluate the aerodynamic lift and drag components of different wing planforms and aircraft architectures with a fast restitution time.

The module considered analytic formulations derived from the theory, literature or from statistical data. Some validation exercises showed that it could be used to estimate the aerodynamic performances of T&W and BWB configurations within an MDAO pre-design process for subsonic cruise flight conditions with a quite satisfactory level of accuracy. The lift slope, the total and individual drag components (lift-induced, viscous and wave) estimated by the module were in quite good agreement with the reference data issued from the numerical and experimental data. In addition, the effects of different elements on the performance, such as fuselage, winglets, nacelles and tail surfaces, were well captured by the module. Finally, the performance of the wings using laminar flow technologies (NLF or HLF) could be estimated with a quite good level of accuracy for use in a pre-design phase.

However, it is important to note that this module cannot be used for detailed optimization (airfoil shape, wing twist or camber, nacelle positions). For instance, the module will estimate the same performances for two wings with similar planform and airfoil thickness evolution with span. These fine optimization steps should be considered in the next stage of the aircraft design process, using more advanced numerical methods once the general architecture is defined in the preliminary phase.

**Funding:** The work presented in this paper was funded by the internal research project CICAV funded by ONERA from 2015 to 2019 and by the EU NACOR Project, which has received funding from the Clean Sky 2 Joint Undertaking under the European Union's Horizon 2020 research and innovation program under grant agreement N°. CS2-AIR-GAM-2018-2019-01. Part of the results used in this paper were carried out in the framework of AIRGREEN2 Project, which received funding from the Clean Sky 2 Joint Undertaking under the European Union's Horizon 2020 research and innovation program, grant agreement N°. 807089—REG GAM 2018—H2020-IBA-CS2-GAMS-2017.

**Institutional Review Board Statement:** Not Applicable.

**Informed Consent Statement:** Not Applicable.

**Data Availability Statement:** Experimental and reference CFD data related to the CRM configurations are available at CRM (https://commonresearchmodel.larc.nasa.gov/ (accessed on 21 November 2022)) or at the Drag Prediction Workshop (https://aiaa-dpw.larc.nasa.gov/ (accessed on 21 November 2022)) websites.

**Conflicts of Interest:** The author declares no conflict of interest.

## Nomenclature

| | |
|---|---|
| b | Wing span (m) |
| c | Airfoil chord (m) |
| d.c. | Drag count ($C_D \times 10^4$) |
| t | Airfoil thickness |
| z | Altitude (m or km) |
| P | Pressure (Pa) |
| T | Temperature (K) |
| V | Velocity (m/s) |
| X | Coordinate in the streamwise direction (m) |
| Y | Coordinate in the spanwise direction (m) |
| LoD | Lift over drag ratio |
| M | Mach number |
| $M_{DD}$ | Divergence Mach number |
| $M_{Cr}$ | Critical Mach number |
| N | Number of components |
| Re | Reynolds number |
| FF | Form factor |
| $Q_N$ | Interference factor for nacelles |
| Osw | Oswald factor |
| $S_{WING}$ | Wing area (m²) |
| $S_{REF}$ | Reference area (m²) |

| $C_l$ | Local (airfoil) lift coefficient |
| $C_L$ | Lift coefficient |
| $C_D$ | Drag coefficient |
| $K_A$ | Airfoil Korn factor |
| $X_{Tra}$ | Index for turbulent or laminar flow computations |
| $X_{Lam}$ | Laminar flow extent (%) |
| NDY | Number of wing section subdivisions |
| AMC | Aerodynamic mean chord (m) |

*Greek Symbols*

| $\alpha$ | Angle of attack (°) |
| $\delta$ | Parameter for Oswald factor |
| $\delta_{WLT}$ | Winglet cant angle (°) |
| $\lambda$ | Wing aspect ratio |
| $\rho$ | Air density (kg/m$^3$) |
| $\mu$ | Dynamic viscosity (kg/(ms)) |
| $\varphi$ | Sweep angle (°) |
| $\varepsilon$ | Wing taper ratio |

*Subscripts*

| LE | Leading-edge |
| TE | Trailing-edge |
| FUS | Fuselage |
| WLT | Winglet |
| ENG | Engines |
| NAC1 | Nacelle or Fan |
| NAC2 | Turbine |
| TAIL | Tail surfaces |

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
