# Peer review of "A Fast Aerodynamic Model for Aircraft Multidisciplinary Design and Optimization Process"

_aerospace, doi:10.3390/aerospace10010007_

Round 1

Reviewer 1 Report

The present paper deals with a low fidelity level aerodynamic module developed for an Overall Aircraft Design (OAD) process. All methods and equations employed are comprehensively described, mentioning also their limitations.  Furthermore, validation and verification of the process by comparison with experimental and numerical data of publicly available configurations are of great value for the reader (as well as other research groups working in the same field).

However, please consider the following minor corrections/modifications:

Line 68: the gravity constat should read 9.80665 instead of 9.800665
Line 106: When first mentioning the Korn factor, it would be helpful to at least refer to section 3.5 of the paper, where the Korn factor is described in more detail. Otherwise readers who are not familiar with the Korn factor a priori coul be puzzled.

Figures 19, 29 and 32: Could you please enlarge the diagrams. In the printout at least the line legends are not readable.

Author Response

See file linked

Reviewer 2 Report

The current work describes a comprehensive air-craft aerodynamic model based on simple sub-models. The work looks robust and the results are good. Only draw-back is the simplified-empirical/analytical type models which can result in inaccuracies. Since this has been mentioned by the authors explicitly, it increases the credibility of the work.

1.) Fig 1: nlarge the figure and improve the resolution. This is a very important diagram and the labels for each of the modules need to be readable

2.) Line 46: Please explain what "Low-fidelity" means here. The requirement for low-fidelity needs to be made clear in relation to the high no of aerodynamic points needed ("design points" seem like a better alternative). It is not possible to run "high fidelity" simulations on all the data sets. 

3.) The significance behind this body type needs to be mentioned. Is it a preferred case for prelim design ? Is there any literature survey on the above ? 

4.) Line 432 : what decides acceptability ?

5.) Check grammar: eg: Line 448 "evaluate" instead of "evaluation"

Author Response

See file linked
